# Cannabinoid CB1 Receptor Involvement in the Actions of CBD on Anxiety and Coping Behaviors in Mice

**DOI:** 10.3390/ph15040473

**Published:** 2022-04-13

**Authors:** Amaya Austrich-Olivares, María Salud García-Gutiérrez, Lucía Illescas, Ani Gasparyan, Jorge Manzanares

**Affiliations:** 1Instituto de Neurociencias, Universidad Miguel Hernández-CSIC, Avda de Ramón y Cajal s/n, San Juan de Alicante, PC03550 Alicante, Spain; aaustrich@umh.es (A.A.-O.); maria.ggutierrez@umh.es (M.S.G.-G.); lillescas@umh.es (L.I.); agasparyan@umh.es (A.G.); 2Red de Investigación en Atención Primaria de Adicciones, Instituto de Salud Carlos III, MICINN and FEDER, PC28046 Madrid, Spain; 3Instituto de Investigación, Sanitaria y Biomédica de Alicante (ISABIAL), PC03550 Alicante, Spain

**Keywords:** cannabidiol, anxiety, depression, cannabinoid receptor 1, cannabinoid receptor 2, G-protein-coupled receptor 55, GABA(A) receptor

## Abstract

The anxiolytic and antidepressant properties of cannabidiol (CBD) have been evaluated in several studies. However, the molecular mechanisms involved in these actions remain unclear. A total of 130 male mice were used. CBD’s ability to modulate emotional disturbances (anxiety and depressive-like behaviors) was evaluated at different doses in wild-type (CD1; 10, 20 and 30 mg/kg; i.p.) and knockout (CB1KO, CB2KO; GPR55KO; 20 mg/kg) mice. Moreover, CBD effects (20 mg/kg; i.p.) were evaluated in mice previously treated with the CB1r-antagonist SR141716A (2mg/kg; i.p.). Relative gene expression analyses of Cnr1 and Cnr2, Gpr55 and GABA(A)α2 and γ2 receptor subunits were performed in the amygdala (AMY) and hippocampus (HIPP) of CD1 mice. CBD (10 and 20 mg/kg) showed anxiolytic and antidepressant actions in CD1 mice, being more effective at 20 mg/kg. Its administration did not induce anxiolytic actions in CB1KO mice, contrary to CB2KO and GPR55KO. In all of them, the lack of cannabinoid receptors did not modify the antidepressant activity of CBD. Interestingly, the administration of the CB1r antagonist SR141716A blocked the anxiolytic-like activity of CBD. Real-time PCR studies revealed a significant reduction in Cnr1 and GABA(A)α2 and γ2 gene expression in the HIPP and AMY of CD1 mice treated with CBD. Opposite changes were observed in the Cnr2. Indeed, Gpr55 was increased in the AMY and reduced in the HIPP. CB1r appears to play a relevant role in modulating the anxiolytic actions of CBD. Moreover, this study revealed that CBD also modified the gene expression of GABA(A) subunits α2 and γ2 and CB1r, CB2r and GPR55, in a dose- and brain-region-dependent manner, supporting a multimodal mechanism of action for CBD.

## 1. Introduction

Mood disorders are considered one of the most prevalent psychiatric disorders with a high socioeconomic and health impact. Anxiety and depression are the most common, with an estimated 280 million people suffering from each [1,2,3,4,5,6]. The treatment of both includes complex pharmacological strategies combined with cognitive-behavioral therapies. Selective serotonin reuptake inhibitors (SSRIs) are the most commonly used antidepressant drugs, with significant limitations regarding their therapeutic effectiveness. Indeed, up to 30% of patients with major depression develop treatment resistance to the first-line selected drugs [7,8,9,10,11]. On the other hand, benzodiazepines, the most commonly prescribed anxiolytic drugs, are limited because of their high risk of abuse and adverse effects [12,13]. Thus, the therapeutic limitations in treating these disorders highlight the need to develop new, more effective and safer pharmacological strategies.

In recent years, the endocannabinoid system (ECS) has attracted interest because of its implication in the pathophysiology of neuropsychiatric disorders, including mood disorders [14,15,16,17,18,19]. Thus, the modulation of this system could be an exciting tool for their treatment. Cannabidiol (CBD) is one of the main compounds of the Cannabis sativa plant without properties as a drug of abuse [20]. This drug can interact with more than 65 different targets, such as G-protein-coupled receptor 55 (GPR55), vanilloid receptors (TRPV1), serotonergic receptor 5-HT1A, mu and delta-opioid receptors and peroxisome proliferator-activated receptor gamma (PPAR-γ) [21,22,23,24,25,26,27]. Notably, CBD acts as a non-competitive allosteric modulator of cannabinoid receptor 1 (CB1r) [21,28] and as an inverse agonist of cannabinoid receptor 2 (CB2r) [25]. Several clinical and preclinical studies showed that CBD presents antidepressant, anxiolytic, antipsychotic and neuroprotective actions, presenting an attractive potential therapeutic strategy for treating mood disorders [29,30,31]. In this respect, its possible utility in anxiety and depression has been evaluated in several animal models, with promising results [32,33,34,35,36,37]. The involvement of 5-HT1A receptors in CBD’s mechanism of action has been proposed [32,33,36]. Additional studies revealed the involvement of CB1r in the anxiolytic activity of CBD [38,39,40].

Cumulative evidence supports the role of cannabinoid receptors (CB1 and CB2) in regulating the response to stress, anxiety, depression, schizophrenia and in cognition [41,42,43,44,45,46,47]. Interestingly, both cannabinoid receptors are associated with significant alterations in the expression of anxiolytic-mediated subunits of the GABA(A) receptor [42,48,49] and with the anxiolytic action of benzodiazepines [41,42]. More recently, additional receptors on which endocannabinoids also act, such as the GRP55 receptor, have been associated with the regulation of emotional reactivity [50,51,52] and hippocampal plasticity [53]. Interestingly, previous studies have demonstrated that CBD modifies the gene expression of these targets in animal models of PTSD [54], alcohol consumption [55,56] and spontaneous cannabinoid withdrawal [57], in which CBD showed efficacy.

The present study aimed to characterize the mechanisms by which CBD exhibits its anxiolytic and antidepressant actions, emphasizing CB1r, CB2r and GPR55. In the first part, we evaluated dose-response acute CBD effects in wild-type animals (WT) in a battery of tests for assessing anxiety and coping-like behavior. In the second part, CBD effects were evaluated in genetically modified mice lacking CB1r (CB1KO), CB2r (CB2KO), and GPR55 (GPR55KO) exposed to representative behavioral tests for measuring anxiogenic- and coping-like behaviors. In addition, pharmacological studies using the CB1r-antagonist, SR141716A, were carried out to further clarify the role of CB1r in CBD effects. Finally, gene expression studies were conducted to analyze potential changes in Cnr1, Cnr2, Gpr55 and GABA(A) genes induced by acute CBD administration in WT animals using real-time PCR.

## 2. Results

### 2.1. Behavioral Evaluation of CBD Actions in WT Mice

We first wanted to evaluate the acute anxiolytic and antidepressant-like effects of CBD. For this purpose, we chose well-accepted animal models for assessing anxiety-like behaviors, using the light-dark box (LDB), the elevated plus maze (EPM) test and novelty suppressed feeding (NSFT) test, coping behavior, and tail suspension test (TST) in rodents. These studies were designed to help further characterize the acute effects of CBD in the modulation of anxiety and depressive-like behaviors in mice.

#### 2.1.1. Light-Dark Box Test (LDB)

Mice treated with CBD at 10 and 20 mg/kg doses spent more time in the lighted box than vehicle (VEH)-treated mice. Interestingly, these anxiolytic actions were not observed at a dose of 30 mg/kg (Figure 1A, one-way ANOVA followed by Student–Newman–Keuls test, F(3,39) = 10.124, *p* < 0.001) (*n* = 9–10/group). In addition, no changes were observed in the number of transitions between groups (Figure 1B, one-way ANOVA followed by Student–Newman–Keuls test, F(3,39) = 0.421, *p* = 0.739) (*n* = 9–10/group).

#### 2.1.2. Elevated Plus Maze Test (EPM)

CBD exerted an anxiolytic-like effect after a dose of 10 mg/kg, increasing the percentage of time spent in the open arms compared to controls. Interestingly, a dose of 20 mg/kg induced a more pronounced anxiolytic action than for CBD-10 mg/kg-treated mice (Figure 1C, one-way ANOVA followed by Student–Newman–Keuls test, F(3,38) = 66,908, *p* < 0.001) (*n* = 9–10/group). In contrast, no effect was observed at the highest dose of CBD (30 mg/kg) compared to the VEH group. No differences were observed in the number of transitions between all four groups (Figure 1D, one-way ANOVA followed by Student–Newman–Keuls test, F(3,35) = 2.056, *p* = 0.126) (*n* = 9–10/group).

#### 2.1.3. Tail Suspension Test (TST)

Treatment with CBD significantly reduced the immobility time at a dose of 20 mg/kg. Interestingly, the lower and the higher doses of CBD (10 and 30 mg/kg) did not induce any effects (Figure 2, One-way ANOVA followed by Student–Newman–Keuls test, F(3,38) = 3.364, *p* = 0.029) (*n* = 9–10/group).

#### 2.1.4. Novelty Suppressed Feeding Test (NSFT)

Mice showed significantly shorter latency time and increased consumption of food pellets (mg) with a dose of 10 mg/kg of CBD compared with the control group. The 20 mg/kg intermediate dose revealed major anxiolytic and hedonic actions. In contrast, the dose of 30 mg/kg did not induce any differences compared with VEH-treated mice (latency time: Figure 3A, one-way ANOVA followed by Student–Newman–Keuls test, F(3,36) = 19.411, *p* < 0.001; Food consumption: Figure 3B, one-way ANOVA followed by Student–Newman–Keuls test, F(3,37) = 16.840, *p* < 0.001) (*n* = 9–10/group).

### 2.2. Effects of CBD on Anxiety and Coping-like Behaviors in Mice Lacking CB1r, CB2r and GPR55

Considering the role of CB1r, CB2r and GPR55 in modulating emotional reactivity, and that they are proposed targets on which CBD directly or indirectly acts, we aimed to explore their involvement in the anxiolytic and antidepressant-like effects of CBD. For this purpose, we evaluated the effects of CBD in the LDB and the TST in mice lacking the CB1r (CB1KO), CB2r (CB2KO) and GPR55 (GPR55KO) receptors. These results enable an improved understanding of the mechanism of action of CBD.

#### 2.2.1. Light-Dark Box (LDB)

Acute CBD (20 mg/kg) administration did not modify anxiety-like behaviors in CB1KO mice compared to the control group (Figure 4A, two-way ANOVA followed by Student–Newman–Keuls test: genotype F(1,40) = 263.42, *p* < 0.001; treatment F(1,40) = 12.386, *p* = 0.001; genotype − treatment: F(1,40) = 13.242, *p* < 0.001) (*n* = 10–11). In contrast, in CB2KO and GPR55KO mice an anxiolytic effect was observed after CBD administration (CB2KO: Figure 4C, two-way ANOVA followed by Student–Newman–Keuls test: genotype F(1,38) = 180.23, *p* < 0.001; treatment F(1,40) = 56.126, *p* < 0.001; genotype − treatment: F(1,40) = 4.829, *p* = 0.035) (*n* = 9–10/group); GPR55KO: Figure 4E, two-way ANOVA followed by Student–Newman–Keuls test: genotype F(1,37) = 23.265, *p* < 0.001; treatment F(1,37) = 53.020, *p* < 0.001; genotype − treatment: F(1,37) = 1.555, *p* = 0.221) (*n* = 8–10/group).

No changes were observed in the number of transitions between CBD- and VEH-treated mice (CB1KO: Figure 4B, two-way ANOVA followed by Student–Newman–Keuls test: genotype F(1,40) = 39.216, *p* < 0.001; treatment F(1,40) = 0.580, *p* = 0.451; genotype x treatment F(1,40) = 0.407, *p* = 0.528) (CB2KO: Figure 4D, two-way ANOVA followed by Student–Newman–Keuls test: genotype F(1,38) = 6.762, *p* = 0.014; treatment F(1,38) = 0.899, *p* = 0.349; genotype x treatment F(1,38) = 1.146, *p* = 0.292) (GPR55KO: Figure 4E, two-way ANOVA followed by Student–Newman–Keuls test: genotype F(1,37) = 24.433, *p* < 0.001; treatment F(1,37) = 0.429, *p* = 0.517; genotype x treatment F(1,37) = 0.306, *p* = 0.584).

#### 2.2.2. Tail Suspension Test (TST)

CBD at the dose of 20 mg/kg elicited antidepressant-like effects among all CB1KO, CB2KO and GPR55KO mice (CB1KO: Figure 5A, two-way ANOVA followed by Student–Newman–Keuls test: genotype F(1,40) = 9.878, *p* = 0.003; treatment F(1,40) = 23.176, *p* < 0.001; genotype x treatment F(1,40) = 0.390, *p* = 0.536; *n* = 10–11) (CB2KO: Figure 5B, two-way ANOVA followed by Student–Newman–Keuls test: genotype F(1,38) = 22.938, *p* < 0.001; treatment F(1,38) = 17.379, *p* < 0.001; genotype x treatment F(1,38) = 0.278, *p* = 0.601; *n* = 9–10) (GPR55KO: Figure 5C, two-way ANOVA followed by Student–Newman–Keuls test: genotype F(1,37) = 16.233, *p* < 0.001; treatment F(1,37) = 14.633, *p* < 0.001; genotype x treatment F(1,37) = 0.0389, *p* = 0.845; *n* = 8–9).

### 2.3. Effects of CBD in Combination with a Selective CB1r Antagonist on Anxiety-like Behaviors in WT Mice

Considering that CBD did not show any anxiolytic-like effect in CB1KO mice, we thoroughly explored the role of CB1r in CBD properties by administering the CB1r-antagonist SR141716A before CBD administration in CD1 mice and evaluated its effects in the LBD test. The results would demonstrate the involvement of CB1r in CBD anxiolytic properties.

The administration of SR141716A did not modify the time spent in the lighted box in the LDB paradigm. In contrast, CBD showed an anxiolytic action at 20 mg/kg. Interestingly, this effect was completely blocked when combined with the CB1r-antagonist, inducing even a mild anxiogenic effect (Figure 6A, two-way ANOVA followed by Student–Newman–Keuls test, SR: (1,31) F = 22,002, *p* < 0.001; CBD: (1,31) F = 0.0427, *p* = 0.838; SR × CBD: (1,31) F = 10,226, *p* = 0.003) (*n* = 8/group). No differences were observed in the number of transitions between groups (Figure 6B, two-way ANOVA followed by Student–Newman–Keuls test, SR: F(1,31) = 2.080, *p* = 0.160; CBD: F(1,31) = 0.382, *p* = 0.542; SR × CBD: (1,31) F = 0.308, *p* = 0.584) (*n* = 8/group).

### 2.4. Gene Expression Studies of Cnr1, Cnr2, Gpr55 and GABA (A)α2 and γ2 Subunits in Wt Mice Treated with CBD

Additionally, we carried out gene expression studies to identify alterations in key targets closely related to emotional reactivity and anxiety, such as Cnr1, Cnr2, Gpr55 and the α2 and γ2 subunits of GABA(A) receptors in the amygdala (AMY) and hippocampus (HIPP) of mice treated with CBD, and two brain corticolimbic regions involved in a broad range of behavioral and cognitive functions, including emotional regulation. We chose real-time PCR to measure these targets’ relative gene expression.

#### 2.4.1. Cannabinoid Receptors

CBD administration induced a dose-dependent decrease in Cnr1 gene expression in the AMY at all doses (Figure 7A, one-way ANOVA followed by Student–Newman–Keuls test, F(3,35) = 6.699, *p* < 0.001) (*n* = 9–10/group). The same effect was observed in the HIPP but only at the highest dose of CBD (Figure 7D, one-way ANOVA followed by Student–Newman–Keuls test, F(3,36) = 4.692, *p* = 0.008) (*n* = 9–10/group). This reduction was accompanied by an increase in gene expression of Cnr2 (at all 3 doses) in the AMY (Figure 7B, one-way ANOVA followed by Student–Newman–Keuls test, F(3,34) = 8.910, *p* < 0.001) (*n* = 9–10/group) and HIPP (at 20 and 30 mg/kg) (Figure 7E, one-way ANOVA followed by Student–Newman–Keuls test, F(3,36) = 7.178, *p* < 0.001) (*n* = 9–10/group). Gpr55 only increased at a dose of 30 mg/kg (Figure 7C, one-way ANOVA followed by Student–Newman–Keuls test, F(3,35) = 3.521, *p* = 0.026) (*n* = 9–10/group) in the AMY, whereas in the HIPP, a significant reduction was observed at all doses tested (Figure 7F, one-way ANOVA followed by Student–Newman–Keuls test, F(3,36) = 4.948, *p* = 0.006) (*n* = 9–10/group).

#### 2.4.2. GABA (A) Receptor Subunits

CBD administration at doses of 20 and 30 mg/kg decreased GABA(A)α2 gene expression in the AMY, this reduction being more pronounced with the highest dose of 30 mg/kg (Figure 8A, one-way ANOVA followed by Student–Newman–Keuls test, F(3,35) = 7.048, *p* < 0.001) (*n*= 9–10/group). Similarly, in the HIPP, a reduction of GABA(A)α2 was observed at all doses of CBD tested (Figure 8C, one-way ANOVA followed by Student–Newman–Keuls test, F(3,36) = 6.759, *p* < 0.001) (*n* = 9–10/group). In addition, for GABA(A)γ2 gene a dose-dependent decrease was observed in both regions, AMY (Figure 8B, one-way ANOVA followed by Student–Newman–Keuls test, F(3,35) = 6.899, *p* < 0.001) (*n* = 9–10/group) and HIPP (Figure 8D, one-way ANOVA followed by Student–Newman–Keuls test, F(3,36) = 4.460, *p* = 0.010) (*n* = 9–10/group).

## 3. Discussion

The present study results confirm that CBD may significantly promote anxiolytic- and antidepressant-like effects in mice in a dose-dependent manner, effects that are mediated, at least in part, by CB1r. This statement is based on the following observations: (1) Low and intermediate acute doses of CBD (10 and 20 mg/kg) induced anxiolytic- and antidepressant-like effects in the behavioral tests assessed in WT mice; (2) Acute CBD administration (20 mg/kg) failed to induce any anxiolytic-like effects in CB1KO mice, whereas it was observed in CB2KO and GPR55KO mice; (3) the administration of the CB1r-antagonist, SR141716A, blocked the anxiolytic action of CBD; (4) CBD presented an antidepressant-like effect in all the knockout mice used; and (4) the administration of CBD reduced Cnr1, GABA(A)α2 and GABA(A)γ2 gene expression in the AMY and HIPP, whereas it increased Cnr2 in both regions. In contrast, Gpr55 gene expression increased in the AMY but decreased in the HIPP after administration of CBD.

Previous studies have shown that CBD induced anxiolytic- and antidepressant-like effects following an inverted U-shape curve, being effective at intermediate but not at very low or high doses [31,58,59,60]. In agreement with these studies, we found that low (10 mg/kg) and intermediate doses (20 mg/kg) induced an anxiolytic-like effect, since both doses increased the exploration time in anxiogenic environments in the LDB and EPM test. Moreover, CBD at these doses also reduced the latency in the NSFT, increasing the food intake. The intermediate CBD dose presented a more robust antidepressant-like effect than the lower dose since it significantly reduced coping behavior, indirectly measured by the immobility time in the TST.

In none of the behavioral tests performed, did CBD at the highest dose (30 mg/kg) show any anxiolytic or antidepressant-like effect, as described previously [61]. However, opposite results were found in other studies in which the same dose induced an anxiolytic- [34,62] or antidepressant-like effect [35,36,63,64]. These discrepancies may be due to differences in methodological procedures, such as different animal species (rats and mice), strains (Wistar rats, C57Bl6J, ICR), behavioral tests applied, and/or the pattern of administration (for more details, see [30]).

Despite evidence supporting the anxiolytic and antidepressant properties of CBD under certain experimental conditions, the complete characterization of the underlying mechanisms of action is still pending. In this respect, 5-HT1A is one of the main targets studied among the more than 65 targets on which CBD acts [29,65,66], demonstrating the involvement of this receptor in its anxiolytic [32,67,68,69,70] and antidepressant-like effects [33,36,40]. Here, we aimed to explore further the implication of additional CBD proposed targets, such as CB1r, CB2r and GPR55, given the critical role these receptors play in emotional reactivity, anxiety and mood disorders [14,42,44,48,51,52,71,72,73]. Studies on genetically modified mice have provided evidence, since CB1KO [47,74,75,76] and CB2KO mice [45,77] showed increased anxiety and depressive-like behaviors. Moreover, recent studies carried out by our group demonstrated that GPR55KO mice also displayed anxiogenic-like responses (to be published).

Genetic (CB1KO) and pharmacological (SR141716A) approaches show CB1r as an undoubtedly active receptor mediating the anxiolytic properties of CBD. Thus, we evaluated the effects of the effective CBD dose (20 mg/kg) in CB1KO, CB2KO and GPR55KO mice. CBD induced anxiolytic-like effects in CB2KO and GRP55KO but did not affect CB1KO mice. Moreover, a pharmacological study using the CB1r-antagonist SR141716A demonstrated that the blockade of CB1r avoids CBD-induced anxiolytic-like effects in the LBD test. Importantly, SR141716A did not induce any effect when it was given alone. These results agree with previous studies demonstrating the involvement of CB1r in the anxiolytic actions of CBD [40,78,79,80,81]. However, when evaluating the effects of CBD on coping behaviors in the tail suspension test in the different knockout mice, the lack of these receptors did not prevent CBD antidepressant effects. Therefore, based on these results, it is tempting to speculate that other receptors, such as the 5-HT1A receptor described above, may be even more critical in understanding the antidepressant action of CBD.

Real-time PCR analyses revealed that acute CBD administration modified gene expression of Cnr1, Cnr2 and Gpr55 in a dose- and brain-region-dependent manner. AMY and HIPP analyses showed that CBD downregulated Cnr1 and increased Cnr2 gene expression dose-dependently, with the most pronounced effects occurring with the highest dose (30 mg/kg). These results agree with previous studies of our group and others demonstrating that CBD treatment reduced Cnr1 [56,82] and increased Cnr2 gene expression in different brain areas [56]. These alterations are compatible with CBD acting as a CB1r-agonist (directly or indirectly) and as a CB2r -antagonist.

Regarding Gpr55, opposite results were observed in the AMY and HIPP. On the one hand, CBD significantly upregulated Gpr55 at the highest dose in the AMY. On the other hand, Gpr55 expression was significantly reduced in the HIPP at all doses tested, with no differences between them. Similarly, our previous studies revealed that CBD reduced Gpr55 in the NAcc of mice exposed to the oral ethanol self-administration paradigm [56]. The exact mechanism by which CBD induced these opposite changes in Gpr55 gene expression between the two brain regions needs to be further explored.

The GABAergic system plays an essential role in the regulation of emotional responses. It is a crucial therapeutic target for controlling anxiety and mood disorders and the critical target by which benzodiazepines (BZD) exert their anxiolytic properties [82]. The anxiolytic effect of BZD is mediated by GABA(A) receptors containing α2 and γ2 subunits, with high expression in the limbic system and cortex [83,84,85,86]. The pentameric GABA(A) receptors are formed by the assembly of different subunits containing α1, α2, α3 or α5, in combination with β and γ2 subunits. Despite less information about the involvement of GABA(A) receptors in depression, studies carried out in patients with major depressive disorder revealed reduced GABA levels, which were normalized after chronic treatment with antidepressants [83,84]. In addition, heterozygous γ2 [85] and α2 [86] knockout mice exhibited more vulnerability to developing anxiety and depressive-like behaviors.

Furthermore, a close interaction between cannabinoid receptors and the GABA system has been demonstrated. Alterations in GABA subunits have been observed in CB1KO knockout mice [49] and mice overexpressing the CB2r (CB2xP) [42]. Interestingly, these mice also showed an impaired anxiolytic action of BZD [41,42]. Pharmacological studies using drugs acting on CB1r and CB2r also showed modified response to stress, anxiety and behavioral despair, and gene expression of GABA(A) subunits, including α2 and γ2 [48,74,87,88]. Thus, considering the role of GABA(A) in anxiety and mood disorders and the crosstalk between GABAergic and cannabinoid systems, we analyzed changes in GABA(A)α2 and GABA(A)γ2 gene expression in the AMY and HIPP of WT mice treated with CBD.

Acute administration of CBD downregulated the gene expression of both GABAergic subunits in the AMY and HIPP at all doses tested. Despite studies of CBD effects in mice lacking α2 and γ2, it would be of great interest to elucidate the exact role of these GABA(A) subunits in CBD anxiolytic properties; it is tempting to speculate that CBD regulates, directly or indirectly, GABA(A) neurotransmission. In line with these findings, previous studies have revealed that CBD inhibited GABA uptake in rat brain synaptosomes at 0.1mM [89]. More recently, an electrophysiological study comparing the actions of CBD and 2-AG on human recombinant GABA(A) receptors expressed on Xenopus oocytes showed that CBD acts as a positive allosteric modulator at GABA(A) receptors containing α2 subunits. This study supported the fact that the site of action of CBD is different from the classic BZD site [90]. Altogether the results obtained suggest that the effects of CBD on GABAergic neurotransmission may be a potential target for its anxiolytic and antidepressant properties that deserve to be explored in future studies.

Overall, the gene expression studies undertaken here further support the complex network through which CBD acts. Behavioral studies revealed that CBD induced anxiolytic- and antidepressant-like effects in WT mice at low and intermediate doses (10 and 20 mg/kg), whereas the highest dose (30 mg/kg) did not induce any behavioral effect. Gene expression studies showed that CBD modified the gene expression of Cnr1, Cnr2 and Gpr55 depending on the doses and the brain region analyzed. Curiously, the highest dose induced the most pronounced changes. Consequently, the study’s main limitation is that gene expression alterations in almost all the targets analyzed were induced by different doses of CBD, making it difficult to correlate some of these biological alterations with the anxiolytic or antidepressant-like effects of CBD. Although future studies are necessary to understand the role of each receptor on CBD actions, based on our results, it is tempting to speculate that the anxiolytic and antidepressant-like effects of CBD may be due to a multimodal mechanism involving different key targets and brain regions, as has been proposed recently [91].

In conclusion, the present study demonstrated that acute administration of CBD produced anxiolytic and antidepressant-like effects in a dose-dependent manner, suggesting that CB1r is one of the crucial targets involved in its anxiolytic properties. Moreover, this study revealed that CBD also modified the gene expression of GABA(A) subunits α2 and γ2 and Cnr1, Cnr2 and Gpr55, in a dose and brain region-dependent manner, indicating that CBD presents a multimodal mechanism of action (Figure 9).

## 4. Materials and Methods

### 4.1. Animals

A total of 130 mice were used in the present study. Forty Swiss CD1 mice were purchased from Charles River Laboratories (Lodi, Italy) to develop the dose-response study with CBD. An additional set of 32 CD1 male mice was used to conduct the pharmacological study with the CB1 receptor antagonist (SR141716A) and CBD. We used twenty-one CB1KO [41,92] and nineteen CB2KO mice [43,44] generated in our laboratory. Dr. Andrei Kolovko kindly provided GPR55KO mice at the Institute of Genomic Medicine. Eighteen of them, bred at our animal vivarium, were used in the present study (TIGM, Houston, TX, USA) [92]. All mice were males and between 2–3 months of age. At the beginning of the experiments, mice were five weeks old and weighed 25–30 g. All animals were maintained under controlled temperature (23 ± 2 °C) and with a light-dark cycle from 0800 to 2000 h, with free access to food (commercial diet for rodents A04 Panlab, Barcelona, Spain) and water. All animal care and experimental studies complied with the Spanish Royal Decree 53/2013, the Spanish Law 32/2007, and the European Union Directive of 22 September 2010 (2010/63/UE), regulating the care of experimental animals and were approved by the Ethics Committee of Miguel Hernández University (ref. UMH.IN.JM.02.17).

### 4.2. Treatment

CBD was obtained from Jazz Pharmaceuticals (Dublin, Ireland) and dissolved in ethanol: cremophor: saline (1:1:18) to obtain the required doses of 10, 20 and 30 mg/kg for wild-type (WT) animals, and the dose of 20 mg/kg for knockout mice (CB1KO, CB2KO and GPR55KO). The drug was prepared immediately before its intraperitoneal (i.p.) administration at a volume of 10 mL/kg of weight (0.3 mL for each mouse). According to its pharmacokinetic properties, CBD was administered 1h and a half before the behavioral evaluation [54,93].

The CB1r-antagonist SR141716A was purchased from Sigma-Aldrich (Madrid, Spain) and dissolved in ethanol, cremophor and saline (1:1:18) to obtain the required dose of 2 mg/kg for its i.p. administration 30 min before CBD administration and 2 h before the behavioral evaluation. The dose of SR141716A was selected based on previous studies demonstrating that this dose does not produce any effects by itself [94,95].

### 4.3. Behavioral Analyses

Mice were randomly divided into groups and subjected to different experimental paradigms to evaluate the anxiolytic and antidepressant actions of the acute administration of CBD. Before every behavioral test, mice were brought to the experimental room in their home cages and were given 60 min to adapt to the environmental conditions of the testing room. The same conditions were maintained for all the behavioral tests. Each test was assessed during the light cycle between 0900 and 1200 h. After each evaluation, mice were undisturbed for 2 to 3 days to allow pharmacokinetic clearance of CBD [93]. WT mice were subjected to a wide range of behavioral evaluations, including the light-dark box (LDB), elevated plus maze (EPM), tail suspension test (TST) and novelty suppressed feeding test (NSFT). According to the results obtained in these studies, only two (LDB and TST) were selected to analyze the emotional behavior in knockout animals (CB1KO, CB2KO and GPR55KO) and CBD’s ability to modulate it. The selective CB1r-antagonist (SR141716A) was used to further evaluate the implication of CB1r in CBD anxiolytic-like effects (20 mg/kg) in the LDB paradigm. For this, WT mice were randomly assigned into four groups: control group (VEH + VEH), CBD group (VEH + CBD), antagonist group (SR-141716A +VEH) or the combination of both drugs (SR-141716A + CBD). The antagonist SR141716 was administered 30 min before CBD, and the behavioral evaluation was carried out 1 h and 30 min after CBD administration.

#### 4.3.1. Light-Dark Box (LDB)

This test uses the natural aversion of rodents to bright areas compared with darker ones [48,96]. The apparatus consisted of two methacrylate boxes (20 × 20 × 15 cm), one transparent and one black and opaque, separated by an opaque tunnel (4 cm). Light from a 60 W desk lamp placed 25 cm above the lightbox provided room illumination. Mice were individually tested in 5 min sessions. At the beginning of the session, mice were placed in the lightbox facing the tunnel that connects to the dark box. The time spent in the lightbox and the number of transitions between the two compartments were recorded in this period. A mouse whose four paws were in the new box was considered to have changed boxes. The apparatus was cleaned between sessions with ethanol 70%.

#### 4.3.2. Elevated Plus Maze Test (EPM)

The EPM consists of two open arms and two enclosed horizontal perpendicular arms 50 cm above the floor [48,97]. The junction of four arms formed a central squared platform (5 × 5 cm). The test began with the animal being placed in the center of the apparatus facing one of the enclosed arms and allowed to explore freely for 5 min. During this period, the time spent in the open arms (as a percentage of total test time) and the number of entries from open arms to closed arms (and vice versa) were recorded. An arm entry was considered an entry of four paws into the arm. The apparatus was cleaned between sessions with ethanol 70%.

#### 4.3.3. Novelty Suppressed Feeding Test (NSFT)

The NSFT was used to measure anxiety-induced hyponeophagia, which is the inhibition of ingestion and approach to food pellets when exposed to an anxiety-provoking novel environment. The testing apparatus consisted of a square, transparent methacrylate cage 40 × 40 × 50 cm, with a food pellet on the white platform in the center of the cage [44,98]. Before the experiment, mice were deprived of food for 24 h, and then each mouse was placed in the corner of the apparatus. The latency time before the mouse started to eat the pellet was recorded up to 5 min. Once the mice began to eat, the total amount of pellets was measured over 5 min. The decrease or increase in the latency time indicates anxiolytic or anxiogenic actions of different drugs, respectively. The anhedonia was measured by calculating the food pellet intake (mg), which increased when mice presented more motivation or ability to experience pleasure.

#### 4.3.4. Tail Suspension Test (TST)

TST is a widely accepted test to evaluate depressive-like behaviors by measuring the immobility time [44,99]. Mice were individually suspended by the tail at the edge of a lever above the tabletop (the distance to the table surface was 35 cm), affixed with the adhesive tape placed approximately 1–2 cm from the tip of the tail. In this situation, mice develop escape-orientated behaviors interspersed with temporally increasing bouts of immobility. The immobility time was measured for 6 min.

### 4.4. Relative Gene Expression Analyses by Real-Time PCR

Relative gene expression analyses of GABA(A)α2 and γ2 subunits, Cnr1, Cnr2 and Gpr55 in the AMY and HIPP, were carried out in WT mice to assess changes in these targets under anxiety or depressive-like conditions and the ability of CBD to modulate them. Briefly, mice were sacrificed 150 min after the administration of CBD (or vehicle) and brain samples were removed from the skull and frozen at −80 °C. These samples were used to obtain coronal sections (500 μm) of regions of interest in a cryostat (−10 °C) according to Paxinos and Franklin’s atlas [100]. Brain nuclei of interest were microdissected following Palkovit’s method as previously modified by our group [101,102]. Total RNA was extracted from brain micropunches with TRI Reagent (Applied Biosystems, Madrid, Spain) and reverse transcription was carried out to obtain the complementary DNA (cDNA) (4374966, High-Capacity cDNA Reverse Transcription Kit with RNase Inhibitor, Applied Biosystems, Madrid, Spain). To perform the real-time PCR, 6.25 μL of water with DEPC (diethylpyrocarbonate, RNAase inhibitor), 5 μL of the cDNA, 11.25 μL of the TaqmanTM Master Mix (4369514, Applied Biosystems, Madrid, Spain), and 1.25 μL of the corresponding Taqman assay were added in each well (4346907, Applied Biosystems, Madrid, Spain). Quantitative analyses of the relative expression of GABA(A) α2 (Mm00433435_m1) and γ2 subunits (Mm00433489_m1), Cnr1 (Mn00432621_s1), Cnr2 (Mm00438286_m1) and Gpr55 (Mm02621622_s1) genes were performed on the StepOne Sequence Detector System (Applied Biosystems, Madrid, Spain). All reagents were used following the manufacturer’s instructions. The reference gene used was 18S rRNA (Mm03928990_g1), and data for each target was normalized to the endogenous reference gene. The fold change in target gene expression was calculated using the 2ΔΔ−Ct method [103].

### 4.5. Data and Statistical Analysis

Statistical analyses were performed using one-way or two-way analysis of variance (ANOVA) followed by the Student–Newman–Keuls post hoc test for comparing four groups affected by the treatment with CBD (and vehicle). Moreover, for the behavioral assay with the antagonist SR141716A, a two-way analysis of variance (ANOVA) followed by the Student–Newman–Keuls post hoc test was assessed for comparing four groups affected by the individual treatment with CBD, SR141716A or its combination. Differences were considered significant if the probability of error was less than 5%. SigmaPlot 11 software (Systat Software Inc., Chicago, IL, USA) was used.

## 5. Conclusions

CBD induced anxiolytic- and antidepressant-like effects in a dose-dependent manner, the intermediate dose (20 mg/kg) being the one that produced these effects most robustly. CB1r appears to be an essential key target for CBD anxiolytic properties.

Changes in Cnr1, Cnr2, Gpr55, GABA(A) subunits α2 and γ2 in limbic areas, including the AMY and HIPP, also suggest that these targets may contribute to CBD effects. Further studies are necessary to understand the specific role of each target and brain region on CBD anxiolytic and antidepressant properties.

## Figures and Tables

**Figure 1 pharmaceuticals-15-00473-f001:**
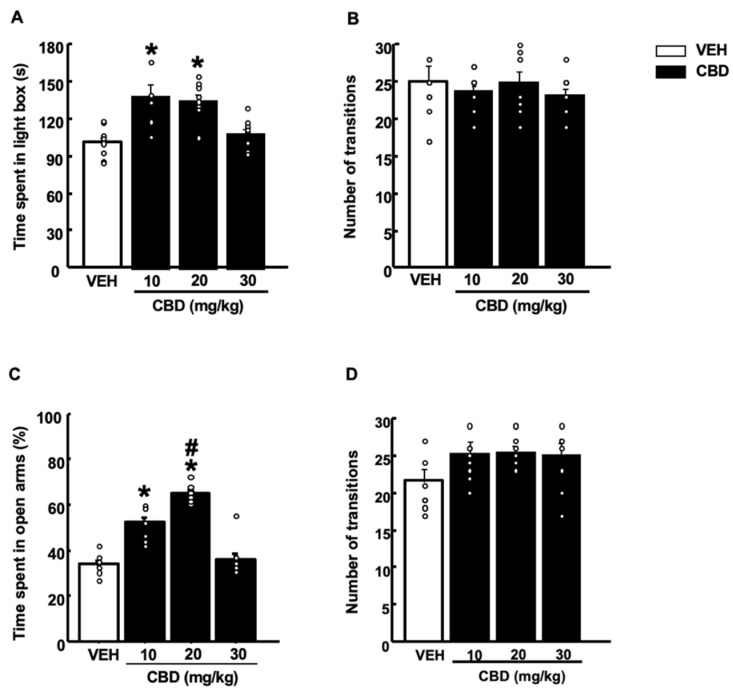
Effects of a single administration of cannabidiol (CBD) (10, 20 and 30 mg/kg, i.p.) on anxiety-like behaviors in the light-dark box (**A**,**B**) and elevated plus maze (**C**,**D**) paradigms. The behavioral evaluation was developed 1 h and 30 min after the administration of CBD (or vehicle (VEH)). Columns represent the means and vertical lines ± SEM of (**A**) the time in the lighted box (s); (**B**) the number of transitions in the light-dark box test; (**C**) the percentage of time in the open arms (%); and (**D**) the number of transitions in the elevated plus-maze test. * Values from CBD-treated mice that were different (*p* < 0.05) from VEH-treated mice and # values from CBD-20 mg/kg-treated mice that were different from CBD-10 mg/kg-treated mice (one-way ANOVA followed by Student–Newman–Keuls test, *p* < 0.05).

**Figure 2 pharmaceuticals-15-00473-f002:**
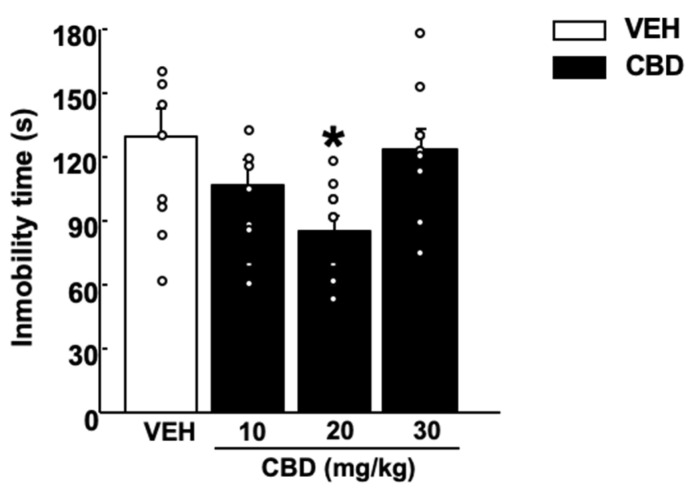
Effects of a single administration of cannabidiol (CBD) (10, 20 and 30 mg/kg, i.p.) on coping behaviors in the tail suspension test paradigm. The behavioral evaluation was developed 1 h and 30 min after the administration of CBD (or vehicle (VEH)). Columns represent the means and vertical lines ± SEM of immobility time (s). * Values from CBD-20 mg/kg-treated mice that were different (*p* < 0.05) from VEH-treated group (one-way ANOVA followed by Student–Newman–Keuls test, *p* < 0.05).

**Figure 3 pharmaceuticals-15-00473-f003:**
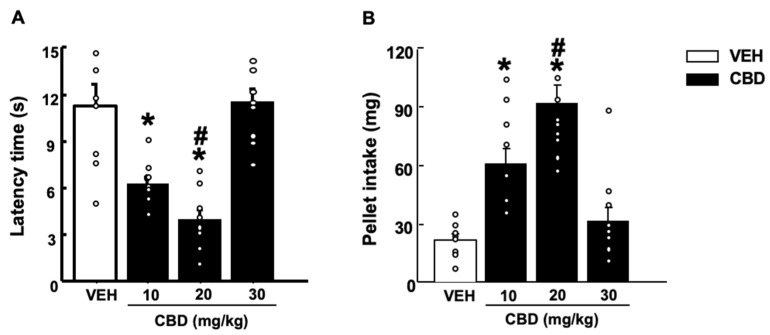
Effect of a single administration of cannabidiol (CBD) (10, 20 and 30 mg/kg, i.p.) on anxiety-like behaviors in the novelty suppressed feeding test. The behavioral evaluation was developed 1 h and 30 min after the administration of CBD (or vehicle (VEH)). Columns represent the means and vertical lines ± SEM of (**A**) latency time (s) and (**B**) pellet intake (mg). * Values from CBD-treated mice that were different (*p* < 0.05) from VEH-treated mice, and # values from 20 mg/kg of CBD-treated mice that were different from mice treated with the lower dose of CBD (10 mg/kg) (one-way ANOVA followed by Student–Newman–Keuls test, *p* < 0.05).

**Figure 4 pharmaceuticals-15-00473-f004:**
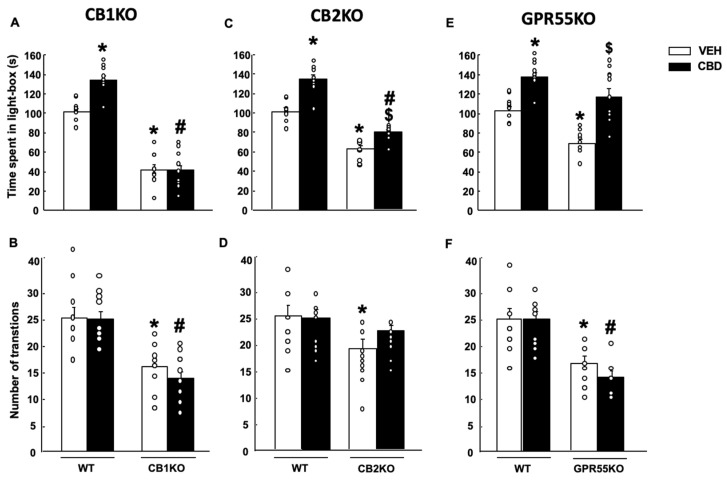
Effects of a single administration of cannabidiol (CBD) at a dose of 20 mg/kg (i.p.) on anxiety-like behaviors in the light-dark box in mice lacking the cannabinoid receptor 1 (CB1KO) (**A**,**B**), lacking the cannabinoid receptor 2 (CB2KO) (**C**,**D**), and lacking the G-protein-coupled receptor 55 (GPR55KO) (**E**,**F**) mice. The behavioral evaluation was developed 1 h and 30 min after the administration of CBD (or vehicle (VEH)). Columns represent the means and vertical lines ± SEM of the time in the lighted box (s) (**A**,**C**,**E**) and the number of transitions (**B**,**D**,**F**). Results from CD1 VEH and CBD (20 mg/kg) groups have been included for comparative purposes. * Values from groups that were different from wild-type (WT)-VEH treated mice (two-way ANOVA followed by Student–Newman–Keuls test, *p* < 0.05). ^$^ Values from groups that were different from KO-VEH treated mice (two-way ANOVA followed by Student–Newman–Keuls test, *p* < 0.05). # Values from groups that were different from WT-CBD-treated mice (two-way ANOVA followed by Student–Newman–Keuls test, *p* < 0.05).

**Figure 5 pharmaceuticals-15-00473-f005:**
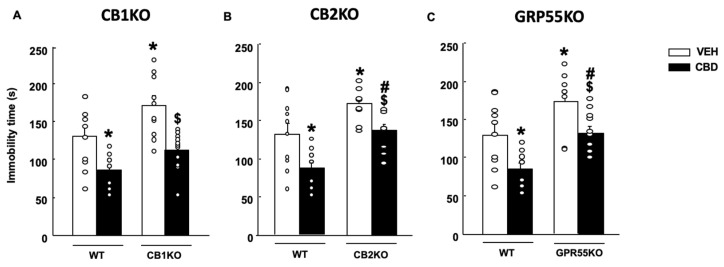
Effects of a single administration of cannabidiol (CBD) at 20 mg/kg (i.p.) on coping behaviors in the tail suspension test in mice lacking the cannabinoid receptor 1 (CB1KO) (**A**), lacking the cannabinoid receptor 2 (CB2KO) (**B**) and lacking the G-protein-coupled receptor 55 (GPR55KO) (**C**). The behavioral evaluation was developed 1 h and 30 min after administration of CBD (or vehicle (VEH)). Columns represent the means and vertical lines ± SEM of immobility time (s). Results from CD1-VEH and CBD (20 mg/kg) groups have been included for comparative purposes. * Values from groups that were different from wild-type (WT)-VEH-treated mice (two-way ANOVA followed by Student–Newman–Keuls test, *p* < 0.05). ^$^ Values from groups that were different from KO-VEH treated mice (two-way ANOVA followed by Student–Newman–Keuls test, *p* < 0.05). # Values from groups that were different from WT CBD-treated mice (two-way ANOVA followed by Student–Newman–Keuls test, *p* < 0.05).

**Figure 6 pharmaceuticals-15-00473-f006:**
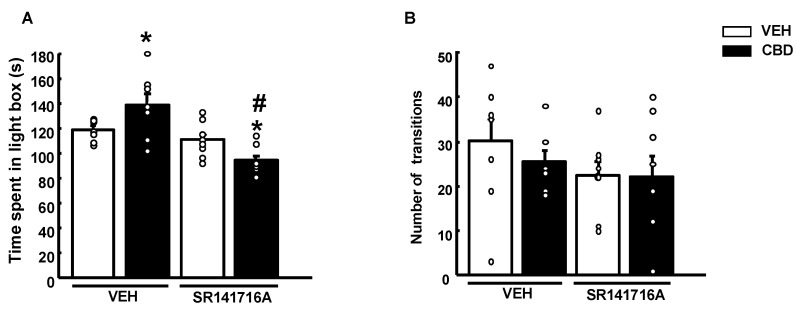
Effects of a single administration of cannabidiol (CBD) (20 mg/kg, i.p.) in mice pre-treated with the cannabinoid receptor 1 (CB1r)-antagonist SR141716A (2 mg/kg, i.p.) on the light-dark box test. The behavioral evaluation was developed 2 h after SR141716A (or vehicle) and 1 h and 30 min after the administration of CBD (or vehicle (VEH)). Columns represent the means and vertical lines ± SEM of the time in the lightbox (s) (**A**) and the number of transitions (**B**). * Values from CBD-treated mice that were different (*p* < 0.05) from vehicle (VEH)-treated mice, and # values from SR141716A + CBD-treated mice that were different from VEH + CBD and SR141716A + VEH-treated animals (two-way ANOVA followed by Student–Newman–Keuls test, *p* < 0.05).

**Figure 7 pharmaceuticals-15-00473-f007:**
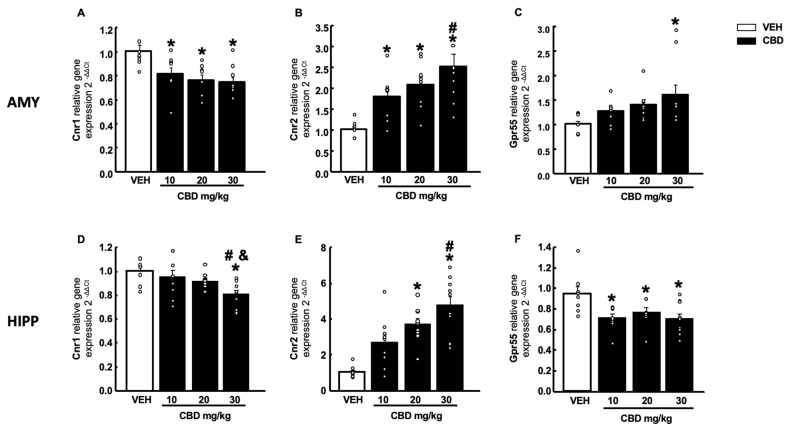
Effects of a single administration of cannabidiol (CBD) (10, 20 and 30 mg/kg, i.p.) on the relative gene expression of cannabinoid receptor 1 (Cnr1), cannabinoid receptor 2 (Cnr2) and G-protein-coupled receptor 55 (GPR55) in the amygdala (AMY) (**A**–**C**) and hippocampus (HIPP) (**D**–**F**). Columns represent the means and vertical lines ± SEM of the relative gene expression (2-ΔΔCt). * Values from CBD-treated mice that were different from vehicle (VEH)-treated mice, and # values from 30 mg/kg of CBD-treated mice that were different from the lower dose of CBD (10 mg/kg) treated animals. & Values from 30 mg/kg of CBD-treated mice that were different from the CBD (20 mg/kg) treated mice (one-way ANOVA followed by Student–Newman–Keuls test, *p* < 0.05).

**Figure 8 pharmaceuticals-15-00473-f008:**
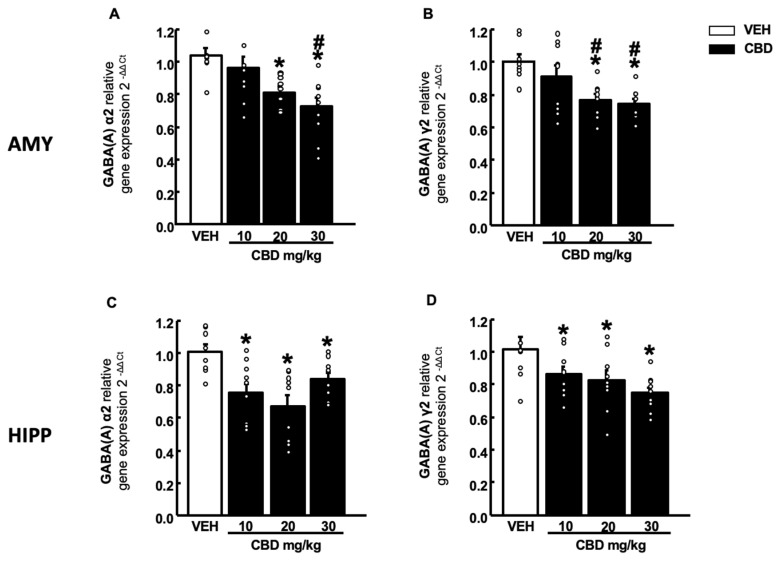
Effects of a single administration of CBD (10, 20 and 30 mg/kg, i.p.) on the relative gene expression of GABA(A) α2 and γ2 in the amygdala (AMY) (**A**,**C**) and hippocampus (HIPP) (**B**,**D**). Columns represent the means and vertical lines ± SEM of the relative gene expression (2-ΔΔCt). * Values from CBD-treated mice that were different from vehicle (VEH)-treated mice and # values from CBD (20 or 30 mg/kg)-treated mice that were different from the lower dose of CBD (10 mg/kg) treated animals (one-way ANOVA followed by Student–Newman–Keuls test, *p* < 0.05).

**Figure 9 pharmaceuticals-15-00473-f009:**
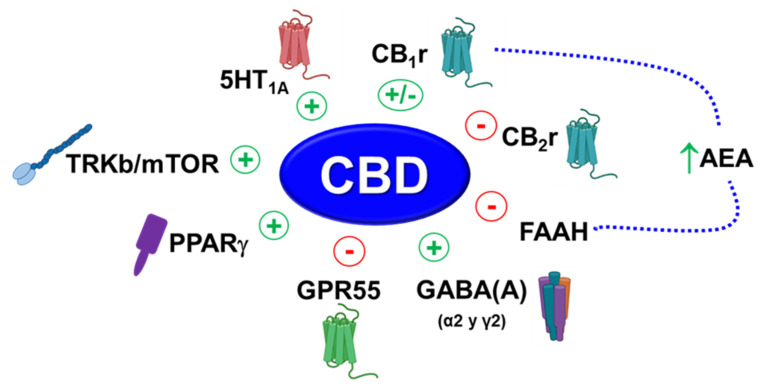
Main targets involved in the anxiolytic and antidepressant actions of CBD. A summary of previously published data and those supported by the present work. 5-HT1A: serotonin 1A receptor; CB1r: cannabinoid 1 receptor; CB2r: cannabinoid 2 receptor; AEA: anandamide; FAAH: fatty acid amide hydrolase; GPR55: G-protein-coupled receptor 55; PPARγ: Peroxisome proliferator-activated receptor gamma; TRKb/mTOR: Tropomyosin receptor kinase B/mammalian target of rapamycin.

## Data Availability

The date is contained within the article.

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
