# Peer review of "Cannabinoid CB1 Receptor Involvement in the Actions of CBD on Anxiety and Coping Behaviors in Mice"

_pharmaceuticals, 2022, doi:10.3390/ph15040473_

Round 1

Reviewer 1 Report

The AA evaulated the molecular mechanisms associated to the anxiolytic and antidepressant actions of CBD. Using genetic and pharmacological approaches, they highlighted the role of CB1 receptors in the anxiolytic effects of CBD.

 there are some concerns that should be addressed:

  • The rationale for studying the gene expression of GABA (A) receptor subunits is not  clear;
  • Fig. 4: the graphical presentation of the WT data appears to be the same in the panels A, C and E

Author Response

Please, find enclosed a copy of the response

Reviewer 2 Report

In this manuscript, Austrich-Olivares et al, provide results on a study to better understand the mechanisms through which CBD exerts its anxiolytic and antidepressant effects, with focus on CB1r, CB2r, and GPR55r receptors. To achieve their goal, they assessed dose-response acute CBD effects in wild-type animals in a battery of anxiety and behavioral despair tests. CBD effects were also assessed in three different knock-out mice each lacking one of the three receptors. These mice were evaluated for anxiogenic and depressogenic-like behaviors. Considering that CBD did not show any anxiolytic-like effect in CB1KO mice, pharmacological studies using the CB1r-antagonist, SR141716A, were conducted to further understand the involvement of CB1r in CBD effects. Results show that targeting CB1r and CB2r is possible to modulate the anxiolytic actions of CBD. Gene expression induced by acute CBD administration in wild-type animals seems to affect these receptors and the anxiolytic-mediated subunits a2 and g2 of GABA(A) receptor in a dose and region-dependent manner (amygdala vs hippocampus).

Minor Revisions

  1. Abstract: In line 27, “AMY” and “HIPP” abbreviations have not been introduced before. The same for the rest of the main text. Please, check along the main text and correct to its first appearance.

  1. Introduction: cannabinoid receptor 1 (Cnr1), cannabinoid receptor 2 (Cnr2) and G-protein coupled receptor 55 (Gpr55), these abbreviations should be introduced in the Introduction section. All justifiable abbreviations should also be included in the figures’ legend.Please revise.

  1. In the results section 2.1: Why do authors describe the novelty suppressed feeding test as an “depressive-like” behaviour? Sure, it is widely used to test anxiety in rodent models of depression but to my knowledge it is more correct to describe it as anxiogenic as, in fact, the authors did in the methods section “anxiety-induced hyponeophagia”. The same for tail suspension test, it is accepted to describe it as depressive-like behaviour but, more recently it is accepted as coping behaviour, like the forced swimming test. Specially because the authors did not perform a stress protocol to model depressive-like behaviour and these experiments are being measured in naïve wild-type animals. Please revise.

  1. Methods Section 4.5 Statistics: Did the authors performed normality tests? Only parametric tests were applied. Do they know the statistical power for the analysis performed? Did the authors consider outliers? Outliers were considered at what Interquartile Range (IQR)? Please, revise to clear statement of the statistical analysis used.

  1. In line 127 of the results section 2.1: Figure legend says “(one-way ANOVA followed by Student test, P< 0.05)”, do the authors refer to Student-Newman-Keuls post-hoc test? If so, please revise all legends to be clearer to the reader. Also, p of Pearson should not be in caps, please correct. Also, include in all figure’s legend the abbreviations of the experimental groups (VEH - vehicle, etc.). Do this to all legends and in the main text.

  1. Figure 4 and 5: The use of two asterisks to differ comparisons between groups is misleading, could you please consider changing this symbol for an $ for instance? I was confused trying to understand if “**” could mean p<0.01, but the legend of this figure does not state that. Please revise.

  1. In section 2.3 of results and 4.2 of methods, please revise to include that the antagonist experiment was performed in wild-type animals only, just to be clearer.

  1. Line 236: “AMY” and “HIPP” abbreviations have not been introduced before.

  1. In Discussion section Line 299: “The present study results support that CBD significantly reduced anxiety- and depressive-like behaviors in mice in a dose-dependent manner, effects mediated, at least in part, by CB1r.”

The authors should be careful when writing “significantly reduced anxiety- and depressive-like behaviors in mice”, specially because these experiments were performed using naïve wild-type animals. Therefore, I would suggest revising the manuscript to avoid words as “reduced”, instead, CBD may promote anxiolytic effect mediated by CBr1. This is suggested by the increase in the exploration time in anxiogenic environments, and increased coping behavior in the TST with reference to the 20mg/kg dosage in naïve wild-type mice. In the CBR1KO animals it was observed the opposite, animals are more anxious and CBD without CBR1 had no action.

  1. Line 308: “increased CB2r” do the authors mean Cnr2 gene expression? Please revise.

Author Response

Please, find enclosed a copy of the response to reviewer's comments

Reviewer 3 Report

The manuscript of Austrich-Olivares et al., studies that the CBD induces anxiolytic- and antidepressant-like effects in a dose-dependent manner, and CB1r appears to be an essential key target for CBD anxiolytic properties.

The manuscript is written in a comprehensible way. However, some sections should be further clarified, revised, or expanded. 
Some specific comments are below:

The introduction section must be expanded.

The methods section is quite complete, although some sub-sections need more detail of the procedures so that they can be reproduced. Improve these sections.

More information in figures would be necessary.

I would recommend adding some summary diagrams or schemes describing in graphical form the anxiolytic and antidepressant properties of CBD regarding your experiments and bibliography.

In vitro biochemical results, that justify or explain the results shown in the work, are missing.

It would be very interesting to see what happens in your model and experiments directly with different cannabinoid ligands such as THC. And, it would also be interesting to see what happens if cannabinoid receptor synthetic agonist/antagonist ligands (WIN, rimonabant…) are used directly, and to show if this is a cannabinoid, CB1R or CB2R-specific effect. Cellular models could be used to start. And, it should be explained/showed if it is described that with some other ligand different all the effects that you see do not occur (negative control).

Cannabinoid receptors can be expressed as dimers, as well as heterodimers with other GPCRs and even with other receptors. Considering the described pharmacological properties of the heteromers, and the potential therapeutic opportunities these complexes may offer, it would be interesting to discuss the possible role of cannabinoid receptor heteromers as pharmacological targets in anxiolytic and antidepressant properties of CBD. All this information is very little covered in the manuscript, and it would be a relevant aspect.

I think the work needs in the discussion deepen into the mechanism of what is happening.

Finally, the overall writing quality of the manuscript is correct, but it’s necessary to correct some spelling mistakes in the text.

Author Response

We have explained in the cover letter to editor the reasons why we will not respond to this reviewer. 

Reviewer 4 Report

CBD, has become one of the interesting pharmaceutical compounds, after its recent FDA approval to treat Epilepsy. However, the mechanism of action of CBD still remains elusive due to its complicated pharmacological profile.  In the current research article, the authors aim to provide mechanistic insights into the CBD’s role as anxiolytic and antidepressant agent. The authors have used mouse models for studying depression and anxiety using Light-dark box test, elevated plus maze test, Tail suspension test (TST) and Novelty suppressed feeding test (NSFT). The authors choose to study the effects of CBD on receptors CB1, CB2, and GPR55. Using a combination of knock out mouse and inverse agonists authors propose that CBD acts through CB1 for its anxiolytic and antidepressant activity. In addition, the authors also find that CBD can modify the expression profiles of GABA, CB1, CB2 and GPR55.

Over all the study is properly designed and data seem to support authors claims. However, I

would like to authors address the following points which might be important to support the hypothesis before its publication in pharmaceuticals.

  1. Authors decision to focus on only 3 receptors, CB1, CB2 and GPR55 for the mouse experiments is not clear. Also, since 5-HT1A has been shown to be involved in multiple studies, why was 5-HT1A not chosen and how does the current targets compare with 5-HT1A in terms of the efficacy of CBD?

  1. The 30mg/kg CBD condition had no effect. The authors need to explain why the effect of CBD diminishes at 30mg/kg in both dark box test and elevated plus maze test.

  1. All the knockouts (CB1, CB2 and GPR55) have significant effect by itself (Increased anxiety-like behaviours). Does that mean, all these receptors are associated with anxiety and depression? Also, since CBD acts as a Negative allosteric modulator/inverse agonist on all these receptors how does that affect the overall anxiolytic and antidepressant profile in these animal models?

  1. For the Light-dark box test and elevated plus maze test the outcome from time spent data almost always does not correlate with the number of transitions. The authors need to explain the importance of these parameters and why they do not correlate in the current study

  1. The knockout experiments were suggestive of the importance of CB1 only in LDB test. However, in Tail suspension test, all the knock out where still responsive to CBD. The authors need to explain this contradiction.

  1. In section 2.3 ( Figure 6) the authors try to further prove the importance of CB1 in CBD action using the CB1 specific inverse agonist SR141716A. However, CBD is known to be negative allosteric modulator and SR141716A is a potent inverse agonist (Ki = ~9nM), which means, both are capable of inactivating CB1 (from the literature it is clear that CBD is much weaker compared to SR141716A). The authors need to explain why a potent inhibitor like SR141716A is unable to have the same anxiolytic and antidepressant if CB1 is an important target.

  1. Can the authors comment on the poor solubility of CBD and its availability to the brain tissues?

  1. The authors claim CB1r as a possible target for CBD for its anti-depression effect. Previously, it is known that CB1 antagonists have an increased risk of anxiety, depression, and suicidality (Bernard et al., 2009). Mechanistically both CBD and SR141716A have the same effect (inhibition) on CB1r. Hence the authors need to explain how CBD manages to provide anxiolytic and antidepressant effects while inhibiting CB1r.

Author Response

(The authors gave the same response as above.)

Round 2

Reviewer 3 Report

I already replied directly to the editor.

Author Response

Dear reviewer, 

Please, find enclosed a point-by-point response to your comments. 

Best regards, 

Jorge Manzanares 

Reviewer 4 Report

The author's response to the comments where overall satisfactory. I have one minor modification/ suggestion (text) which might be helpful in providing the most insights.

Most of the comments were made since the title of the article is “Molecular mechanisms associated to the anxiolytic and antidepressant actions of cannabidiol”. However, based on reading the article and the author's response it is clear that the authors have tried to study the effects of CBD on specific receptors using anxiolytic and antidepressant animal models. Since CBD has multiple targets, as authors agree, along with several supporting research and along with the fact that it is still unknown, why a strong inverse agonist like SR141716A is unable to mimic the effect of CBD which is a negative allosteric modulator of CB1, it might be premature to propose a CB1 mediated mechanism for cannabidiol.

Additionally, the authors also agree that CBD pharmacological results are controversial since different groups have different efficacy and target data.

Hence, it would be more accurate if the title of the article can be more specific, which indicates that CBD can rescue anxiolytic and antidepressant animal models in CB1 KO mouse or CBD can modify multiple targets rather than calling it a generic mechanism of action.

Author Response

Dear reviewer

Please, find enclosed a point-by-point response to your comments.

Best regards,

Jorge Manzanares

This manuscript is a resubmission of an earlier submission. The following is a list of the peer review reports and author responses from that submission.

Round 1

Reviewer 1 Report

The authors wanted to investigate the molecular mechanisms associated to the anxiolytic and  antidepressant actions of cannabidiol by using KO mice for cannabinoid receptors and GPR55 receptors. The paper is well written and deals with a very current topic, although the conclusion that CBD induces anxiolytic and antidepressant effects by engaging CB1 receptors is not adequately supported by the obtained results. In my opinion, all behavioral analyses, in which there is a significant effect from CBD treatment, should be performed in KO CB1. In addition, a correlation study between molecular data (gene expression) and clinical data (behavioral analyses) is mandatory.

I have some comments for the authors:

  1. Experiments of percentage of time in the open arms as well as latency time and pellet intake related to food consumption analysis should also repeat in KO mice.
  2. The authors should consider evaluating CB1, CB2 and GPR55 gene expressions in CBD-treated mice also in the hippocampus as they did for GABA receptors.
  3. I think that the gene expression data of the GABA receptors are not useful unless the authors intend to evaluate the behavioral studies in KO mice for the GABA receptors.
  4. The authors cite several times the involvement of serotonergic receptors in the action of CBD. The authors could use mice KO for HT1A and compare them with CB1 KO mice treated with CBD.
  5. A typo is present on page 5, line 141: “increase” instead of “decrease”.

Author Response

Comments and Suggestions for Authors

The authors wanted to investigate the molecular mechanisms associated to the anxiolytic and antidepressant actions of cannabidiol by using KO mice for cannabinoid receptors and GPR55 receptors. The paper is well written and deals with a very current topic, although the conclusion that CBD induces anxiolytic and antidepressant effects by engaging CB1 receptors is not adequately supported by the obtained results. In my opinion, all behavioral analyses, in which there is a significant effect from CBD treatment, should be performed in KO CB1.

Resp: Thank you very much for your kind comments. Despite its interest, we did not consider it necessary to perform all the behavioral assays in CB1KO mice. We evaluated the effects of CBD in knockout mice using one representative test (LBD and TST) for each set of paradigms used for studying the anxiolytic (LBD and EPM) and antidepressant properties (TST and NSFT) of CBD in WT mice. Interestingly, we have carried out an additional study to strengthen the role of CB1r in the anxiolytic-like effects of CBD. Briefly, the CB1r-antagonist SR141716A have been administered before CBD evaluating CBD anxiolytic properties in the LDB. The results revealed that SR141716A blocked CBD effects. Importanly, SR141716A did not induce any effect when given alone. Thus, this new experiment contribute to further support the involvement of CB1r in in the anxiolytic properties of CBD.

In addition, a correlation study between molecular data (gene expression) and clinical data (behavioral analyses) is mandatory.

Resp: Thank you very much for the interesting comment. Unfortunately, we cannot correlate behavioral effects and alterations in gene expression. Anxiety-like behavior involves the participation of several targets and many brain areas, therefore resulting often very complicated to find a clear correlation between neurochemical changes and a specific type of behavior. Moreover, the pharmacological mechanisms of action of CBD imply the contribution of more than 60 receptor targets. CBD induced anxiolytic- and antidepressant-like effects in WT mice at low and intermediate doses (10 and 20 mg/kg). However, a high dose (30 mg/kg) did not induce any behavioral effect. Gene expression studies indicated that the administration of CBD differentially modified the gene expression of CB1, CB2 and GPR55 depending on the doses and the brain regions analyzed. Based on the results found, it is tempting to speculate that the mechanism of action of CBD spans different genes and brain regions. Consequently, its efficacy probably lies in modulating other brain circuits, as proposed recently (PMID: 34485973).

To emphasize this point, we have modified the discussion section. Please, see the second last paragraph.

I have some comments for the authors:

  1. Experiments of percentage of time in the open arms as well as latency time and pellet intake related to food consumption analysis should also repeat in KO mice.

Resp: Thanks for the comment. As we explained previously, we selected one representative behavioral test for studies using mice modified genetically to assess anxiolytic- and antidepressant-like effects of CBD. We consider that the results show that CB1r are necessary for CBD to exert its anxiolytic action. Performing the other tests would almost certainly give us the same results. Likewise, we currently do not have enough KO mice to perform these experiments. More importantly, we have carried out an additional study with the  SR141716A that further supports the role of the CB1r in the anxiolytic action of CBD, as we previously mentioned in our above response to the reviewer comment.

  1. The authors should consider evaluating CB1, CB2 and GPR55 gene expressions in CBD-treated mice also in the hippocampus as they did for GABA receptors.

Resp: In agreement with the reviewer’s suggestion, we measured the gene expression of CB1, CB2 and GPR55 receptors in the HIPP of mice treated with CBD. The results revealed that CBD significantly reduced CB1r and Gpr55 in the HIPP at 30 mg/kg and all doses, respectively. In the case of CB2r, we found a significant increase at 20 and 30mg/kg. We have modified different sections of the manuscript accordingly to these new results.

  1. I think that the gene expression data of the GABA receptors are not helpful unless the authors intend to evaluate the behavioral studies in KO mice for the GABA receptors.

Resp: GABAA receptors are crucial targets involved in emotional reactivity. The primary pharmacological approach for treating anxiety is benzodiazepines (BZD). These drugs act as an allosteric modulator of GABAA receptors. In particular, the GABAA a2 and g2 subunits, are more closely related to its anxiolytic properties. Previous studies revealed that CBD inhibited GABA uptake in rat brain synaptosomes at 0.1mM. More recently, an electrophysiological study compared the actions of CBD and 2-AG on human recombinant GABA(A) receptors expressed on Xenopus oocytes. The results revealed that CBD was a positive allosteric modulator at GABAA receptors, finding the maximal level of GABA enhancement on GABAA receptors α2 subunits. This study supported the fact that the site of action of CBD is different from the classic BZD site.

The purpose of our study was to characterize further if the modulation of GABA may be a mechanism of action by which CBD displays its anxiolytic-like effects. We measured gene expression changes of the GABAA subunits a2 and g2 in the amygdala and hippocampus, brain regions of the corticolimbic system related to stress response and emotional reactivity.

The results suggest that CBD regulates GABA(A) neurotransmission, implying its role as a positive allosteric modulator of GABA(A) receptors. Considering the reviewer’s suggestion, we have modified the discussion section highlighting that the evaluation of CBD effects in mice lacking a2 and g2 would be of interest to elucidate the exact role of these GABAA subunits in CBD anxiolytic properties.  

  1. The authors cite several times the involvement of serotonergic receptors in the action of CBD. The authors could use mice KO for HT1A and compare them with CB1 KO mice treated with CBD.

Resp: Previous studies demonstrated the involvement of 5-HT1A receptors in the anxiolytic (e.g., PMID: 25701682, 24321837, 20945065 etc.) and antidepressant properties of CBD (e.g., PMID: 26711860, 20002102, etc.), including our study (PMID: 30324842). Therefore, we do not consider performing these behavioral tests in mice KO for 5-HT1A since its role in CBD actions has already been demonstrated. The main objective of this study was to characterize further the role of additional key targets on which CBD has been shown to act, directly or indirectly, and that could be related to its anxiolytic and antidepressant actions. For this reason, we select the cannabinoid receptors (CB1 and CB2) and the GPR55 (a non-cannabinoid receptor on which cannabinoid ligands act) due to their involvement in emotional reactivity, anxiety and behavioral despair. In its current form, we consider that our study provides evidence supporting CB1r as an additional key target to explain the anxiolytic activity of CBD. The exact contribution of each receptor (CB1r and 5-HT1A) to the anxiolytic effects of CBD should be clarified in future studies.

  1. A typo is present on page 5, line 141: “increase” instead of “decrease”.

Resp: We have corrected the typo accordingly. 

Reviewer 2 Report

Amaya Austrich-Olivares et al. investigated the molecular mechanisms associated to the anxiolytic and antidepressant actions of cannabidiol. Overall, this is an important subject and the authors used appropriate methods to look for such mechanisms, however there are some major and minor problems in this manuscript.

Major issues:

  1. There are two main new findings in this research- A. CB1 seems to be important for the anxiolytic action of CBD. B. CBD treatment affects gene expression of endocannabinoid receptors and GABA(A) receptors in the brain. The authors did not explain the connection between these two main findings, even in the discussion.
  2. The results part should be re-constructed. In each model the results of the KO mice must be compared to the results of the WT. The WT results are the proper controls for the KO experiments. Therefore, the results of fig.1 and fig 4 need to be in the same figure, fig.2 and fig 5 need to be in the same figure and food consumption should be tested in the KO mice. 
  3. In all the KO experiments, the results in the vehicle groups are significantly different from the vehicle groups in the WT. If there is no technical problem that explains this, the meaning of these results could be that the expression of the tested endocannabinoid receptors is important in anxiety- and depressive-like behaviors in mice. This is an important result which should be discussed.

Major issues:

  1. Introduction- CBD is not a ligand for CB1 receptors, but It behaved as a non-competitive negative allosteric modulator of CB1.
  2. Methods- Add references or data about the KO mice. Add mode of administration for CBD (IP, oral ?). Add age and gender of the mice (very important since it affects the results of cannabinoid treatments).
  3. Results- for each part add an introductory sentence that states the main question you address in this part, why it is important and how will the chosen method help you to answer it.
  4. Discussion- explain how do all the results combine to add to our knowledge about the role of the endocannabinoid system in anxiety and depression and the possible effects of CBD on this system.

Author Response

Comments and Suggestions for Authors

Amaya Austrich-Olivares et al. investigated the molecular mechanisms associated to the anxiolytic and antidepressant actions of cannabidiol. Overall, this is an important subject and the authors used appropriate methods to look for such mechanisms, however there are some major and minor problems in this manuscript.

Major issues:

  1. There are two main new findings in this research- A. CB1 seems to be essential for the anxiolytic action of CBD. B. CBD treatment affects gene expression of endocannabinoid receptors and GABA(A) receptors in the brain. The authors did not explain the connection between these two main findings, even in the discussion.

Resp: We appreciated the comment made by the reviewer. We have carried out an additional study to support further the role of CB1r in the anxiolytic action of CBD.

     CBD was evaluated in mice previously treated with the CB1r antagonist, SR141715A. The results revealed that the administration of this receptor antagonist blocked the anxiolytic action of CBD in the LDB. Thus, genetic (CB1KO) and pharmacological studies (SR141716A) pointed out the CB1r as a crucial target to understand the anxiolytic actions of CBD.

     Furthermore, we measured CB1, CB2 and GPR55 gene expressions in the HIPP of WT mice treated with CBD. The results revealed that CBD significantly reduced CB1r and GPR55 in the HIPP at 30 mg/kg and all doses, respectively. Interestingly, CB2r was significantly up-regulated with 20 and 30 mg/kg doses. The results revealed that CBD modified the gene expression of CB1, CB2, GPR55 and the a2 and g2 GABA(A) subunits in the AMY and HIPP of WT mice.

In summary, CBD induced an anxiolytic- and antidepressant-like effect at 10 and 20 mg/kg, whereas the high dose (30 mg/kg) failed to cause any change. The gene expression studies revealed that high doses induce more pronounced changes than the other doses in different targets, for example, in the CB2r and GPR55 in the AMY. Curiously, there was no difference in CB1r gene expression in the AMY between doses. Unfortunately, we cannot correlate behavioral effects and alterations in gene expression. However, it is tempting to speculate that based on the complex CBD pharmacology that spans different genes and brain regions, its efficacy probably lies in modulating other brain circuits, as have been proposed recently (PMID: 34485973). In the discussion of the manuscript, we further discussed this point. Please, see the second last paragraph.

  1. The results part should be re-constructed. In each model, the results of the KO mice must be compared to the results of the WT. The WT results are the proper controls for the KO experiments. Therefore, the results of fig.1 and fig 4 need to be in the same figure, fig.2 and fig 5 need to be in the same figure, and food consumption should be tested in the KO mice. 

Resp: Thanks for the comment. WT and KO mice studies have been carried out in different experiments. Firstly, we conducted studies in WT evaluating acute doses-responses effects of CBD in the battery of behavioral tests (LDB, EPM, TST, NSFT). Once we characterized the anxiolytic and antidepressant acute effects of CBD, we conducted the second part of experiments in KO mice to clarify the role of CB1, CB2 and GPR55 in the observed effects of CBD. Thus, the purpose of studies using KO mice was to demonstrate the role of these receptors in the actions of CBD. Considering these facts, we do not think comparing WT and KO mice is necessary.

As we previously mentioned, we have conducted an additional experiment to clarify further the involvement of CB1r in the anxiolytic-like actions of CBD. We evaluated in KO mice the effects of CBD in LBD and TST, one representative test for each set of paradigms used for studying the anxiolytic (LBD and EPM) and antidepressant properties (TST and NSFT) of CBD in WT mice. The results revealed that the pre-administration of CB1r-antagonist SR141716A significantly blocked the anxiolytic actions of CBD. Altogether, we considered it unnecessary to carry out NSFT in CB1KO mice.

  1. In all the KO experiments, the results in the vehicle groups are significantly different from the vehicle groups in the WT. If there is no technical problem that explains this, the meaning of these results could be that the expression of the tested endocannabinoid receptors is important in anxiety- and depressive-like behaviors in mice. This is an important result which should be discussed.

Resp: Thanks for the comment. Indeed, previous studies by our group and other colleagues demonstrated that these KO mice presented an increased vulnerability to stressful stimuli, anxiety, and depression than WT. Besides, their pharmacological modulation induced anxiolytic or anxiogenic-like effects, depending on the drug used and the doses tested. We have included an additional paragraph in the discussion section to reference previous pharmacological and genetic studies demonstrating that mice lacking these cannabinoid receptors showed increased anxiogenic- and depressogenic-like responses.  

Major issues:

  1. Introduction- CBD is not a ligand for CB1 receptors, but It behaved as a non-competitive negative allosteric modulator of CB1.

Resp: In agreement with the reviewer, we have included that CBD behaves as a non-competitive allosteric modulator of CB1. Please see page 2, lines 46-49.

  1. Methods- Add references or data about the KO mice. Add mode of administration for CBD (IP, oral ?). Add age and gender of the mice (very important since it affects the results of cannabinoid treatments).

Resp: We have included references for KO mice and gender (males) and age (2-3 months) of mice used. Besides, the type of CBD administration (i.p.) was also added.

  1. Results- for each part add an introductory sentence that states the main question you address in this part, why it is important and how will the chosen method help you to answer it.

Resp: In agreement with the reviewer’s comment, we have introduced a sentence before each part of the results.

  1. Discussion- explain how do all the results combine to add to our knowledge about the role of the endocannabinoid system in anxiety and depression and the possible effects of CBD on this system.

Resp: According to the comment made by the reviewer, we have modified the discussion to emphasize the role of the endocannabinoid system in anxiety and depression and how CB1r is more closely related than CB2 and GPR55 in the anxiolytic-like actions of CBD.

Reviewer 3 Report

In this manuscript, the authors investigated the molecular mechanisms associated to the anxiolytic and antidepressant actions of cannabidiol. Specifically, they highlighted the role of CB1 receptors in these effects through the use of knock-out mice.

This is a potentially interesting study, however there are some concerns that should be addressed:

-The introduction is too short and need to be improved with more details regarding the purpose of the paper.

- the discussion focuses more on the role of the GABAergic system in the regulation of emotional responses without arguing the role of the endocannabinoid system in the observed effects.

For example, some studies showed that CBD promotes hippocampal neurogenesis by modulating the endocannabinoid system.

In general, the role of the endocannabinoid system in Anxiety must be discussed.

-In the Materials and Methods, the authors wrote that the Relative gene expression analyses of GABA(A)α2 and É£2 subunits, Cnr1, Cnr2 and 336 Gpr55 were carried out in the amygdala (AMY) and hippocampus (HIPP) of WT mice.

Fig. 6 only show the effects of a single administration of CBD on the relative gene expression of cCnr1, Cnr2 and Gpr55 in amygdala .

-Were both males and females used in the experiments?

-How were the animals selected for gene expression analysis?

-Please, specify the number of the animals used for each experiment

- It is very important to add experiments aimed to verify if the effects of CBD observed in WT would be prevented by the pre-treatment with both CB1 and CB2 antagonists. 

Author Response

In this manuscript, the authors investigated the molecular mechanisms associated to the anxiolytic and antidepressant actions of cannabidiol. Specifically, they highlighted the role of CB1 receptors in these effects through the use of knock-out mice.

This is a potentially interesting study, however there are some concerns that should be addressed:

-The introduction is too short and need to be improved with more details regarding the purpose of the paper.

Resp: According to the reviewer, a new paragraph has been included in the introduction section emphasizing the purpose of the paper.

- the discussion focuses more on the role of the GABAergic system in regulating emotional responses without arguing the role of the endocannabinoid system in the observed effects. For example, some studies showed that CBD promotes hippocampal neurogenesis by modulating the endocannabinoid system. In general, the role of the endocannabinoid system in anxiety must be discussed.

Resp: We have considered the reviewer comment and included an additional paragraph with the role of ECS in anxiety and CBD effects in the discussion section.

-In the Materials and Methods, the authors wrote that the Relative gene expression analyses of GABA(A)α2 and É£2 subunits, Cnr1, Cnr2 and 336 Gpr55 were carried out in the amygdala (AMY) and hippocampus (HIPP) of WT mice. Fig. 6 only shows the effects of a single administration of CBD on the relative gene expression of cCnr1, Cnr2 and Gpr55 in amygdala.

Resp: In the revised version of the manuscript, we have carried out additional studies measuring Cnr1, Cnr2 and Gpr55 gene expressions in the hippocampus.

-Were both males and females used in the experiments?

Resp: We have included information regarding the gender of mice used (males) in the methods section.

-How were the animals selected for gene expression analysis?

Resp: All the mice used in the behavioral assessments were selected for the gene expression studies.

-Please, specify the number of the animals used for each experiment

Resp: Thank you for the comment. We have included the information about the number of animals in the results section.

- It is crucial to add experiments to verify if the effects of CBD observed in WT would be prevented by the pre-treatment with both CB1 and CB2 antagonists. 

Resp: We have considered the suggestion made by the reviewer and carried out an additional study to evaluate the anxiolytic-like actions of CBD in mice pre-treated with the CB1r-antagonist SR141716A in the LDB test. The results revealed that the administration of SR141716A blocked the anxiolytic-like activity of CBD. Interestingly, SR141716A did not induce any effect when it was given alone. These results further support the participation of CB1r in the mechanism involved in the anxiolytic action of CBD.

We have not carried out studies with CB2r antagonists considering that CBD still induced anxiolytic and antidepressant-like effects in CB2KO mice.

Round 2

Reviewer 1 Report

although the authors did not satisfy all the experimental requests, they still adequately commented on the critical points in the revised manuscript and therefore in my opinion it can be accepted for publication

Reviewer 2 Report

Overall, the article was improved. However, there are a serious problems with the controls in the experiments. In order to conclude something from experiments using knock out mice, there should be age and gender matched WT control groups in the same experiment. Otherwise, the differences may be the results of technical issues.

In addition, the control in the new experiment does not repeat the results of fig. 1 (no significant elevation in the time spent in the lighted box). I think that there are interesting results in the manuscript, but it is not ready for publication at this time.

Reviewer 3 Report

The Autors addressed all the points highlighted.

 Please, read the text carefully and correct small errors